

# In silico analyses of CD14 molecule reveal significant evolutionary diversity, potentially associated with speciation and variable immune response in mammals

Olanrewaju B. Morenikeji and Bolaji N. Thomas

Department of Biomedical Sciences, Rochester Institute of Technology, Rochester, NY, USA

## ABSTRACT

The cluster differentiation gene (CD14) is a family of monocyte differentiating genes that works in conjunction with lipopolysaccharide binding protein, forming a complex with TLR4 or LY96 to mediate innate immune response to pathogens. In this paper, we used different computational methods to elucidate the evolution of CD14 gene coding region in 14 mammalian species. Our analyses identified leucine-rich repeats as the only significant domain across the CD14 protein of the 14 species, presenting with frequencies ranging from one to four. Importantly, we found signal peptides located at mutational hotspots demonstrating that this gene is conserved across these species. Out of the 10 selected variants analyzed in this study, only six were predicted to possess significant deleterious effect. Our predicted protein interactome showed a significant varying protein–protein interaction with CD14 protein across the species. This may be important for drug target and therapeutic manipulation for the treatment of many diseases. We conclude that these results contribute to our understanding of the CD14 molecular evolution, which underlays varying species response to complex disease traits.

# INTRODUCTION

The cluster of differentiation 14 (CD14) gene is a surface differentiation antigen preferentially expressed on mammalian monocytes, neutrophils, macrophages, and plasma cells (*Baumann et al., 2010*; *Tang et al., 2017*). It encodes a protein that is important for initiating a robust immune response against microbial pathogens by mediating innate immune response, in concert with several other proteins. It is a co-receptor with Toll-like receptor-4 (TLR4) to activate several intracellular signaling pathways that lead to the synthesis and release of inflammatory cytokines, antimicrobial peptides, chemokines, and other co-stimulatory molecules which in turn interact with the adaptive immune system (*Härtel et al., 2008*). Comparative studies have shown that two or more proteins can have common evolutionary origin thereby sharing structural and functional characteristics (*Kanduc, 2012*). The CD14 molecule exists in two forms: soluble (sCD14) or membrane-bound (mCD14) (*Panaro et al., 2008*; *Xue et al., 2012*). There are multiple variants of the

Corresponding author
Bolaji N. Thomas, bntsbi@rit.edu

CD14 molecule that are encoded by the same protein due to alternative splicing and as such has been mapped to varying chromosomal locations in different species. For example, it is mapped to chromosome 5 in humans, chromosome 7 in cattle and chromosome 18 in mouse (*Ferrero et al., 1990*; *Le Beau et al., 1986*; *Ibeagha-Awemu et al., 2008*).

Studies in human, mouse, cattle, and sheep have shown that CD14 is significantly involved in innate immunity, playing major roles in susceptibility to tuberculosis, trypanosomosis, malaria, and other bacterial infections (*Sugawara et al., 2001*; *Ibeagha-Awemu et al., 2008*; *Xue et al., 2012*; *Ojurongbe et al., 2017*). Other published reports have shown that there is a higher susceptibility to *Mycobacterium tuberculosis* infection in CD14 transgenic mice compared to the wild type (*Reiling et al., 2002*; *Wieland et al., 2008*). Likewise, single nucleotide polymorphisms (SNPs) in CD14 gene have been associated with higher susceptibility in many disease instances (*Oakley et al., 2009*; *Liu et al., 2012*; *Xue et al., 2012*; *Zanoni & Granucci, 2013*; *Xue et al., 2018*). In fact, *Song et al. (2014)* reported how genetic heterozygosity modulate disease resistance and progression in cattle infected with bovine tuberculosis. Furthermore, comparative studies have shown that organism relatedness can be traced through their pattern of genetic divergence (*Kanduc, 2012*; *De Donato et al., 2017*; *Peters et al., 2018*).

Several sequence-based methods and tools have been developed to glean evolutionary information in related species via amino acids sequence variation and conservation of homologous proteins through multiple sequence alignment (MSA) (*Hepp, Gonçalves & De Freitas, 2015*; *Peters et al., 2018*). Similarly, other computational methods are available to identify SNP variation within and between amino acid sequences in multiple species, which possibly affecting the stability and functionality of such proteins (*Ng & Henikoff, 2006*; *Yue & Moult, 2006*; *Hepp, Gonçalves & De Freitas, 2015*). Many of these tools can predict the effect of SNP occurrence in protein sequences to determine whether they are disease related, deleterious or neutral. Comparative genomics therefore is a powerful tool to elucidate variants and effects among multiple species in order to detect diseases associated with variations. Variations in amino acid sequence have the ability to alter protein structure and functions like ligand binding, protein folding, impaired intracellular transport, and reduced stability (*Zeron-Medina et al., 2013*; *Morisseau et al., 2014*; *Valastyan & Lindquist, 2014*).

Due to the significance of the CD14 gene in several disease cases in humans and other species, in addition to its considerable involvement in innate immunity, we hypothesize that there might be evolutionary patterns of similarity and diversification that occurred during speciation, which is important for comparative immune and disease studies in different species. To this end, we carried out a detailed comparative study of CD14 protein in 14 mammalian species to elucidate the evolutionary basis for conserved regions, active sites and mutational hotspots, which could lead to novel disease phenotypes. In addition, we examine the diversification in CD14 protein interactions within and across the species, which could be explored for therapeutic development or drug design.

## MATERIALS AND METHODS

### Sequence retrieval and multiple sequence alignment

Complete CD14 amino acid sequences of 13 mammals were retrieved from UniProtKB/ Swiss-Prot (https://www.uniprot.org/uniprot/?query=CD14&sort=score) database. The sequences were retrieved for human (P08571), rat (Q63691), mouse (P10810), cattle (Q95122), rabbit (Q28680), monkey (B3Y6B8), gorilla (G3R4C0), sheep (W5QJA2), horse (F6VK89), pig (A7BG66), buffalo (A0A2R4SDF9), chimpanzee (B3Y6B4), and yak (L8I9P7). The amino acid sequence for goat (ABE68725.1) was retrieved from GenBank. We performed sequence alignment with the Multalin software (http://multalin.toulouse.inra.fr/multalin/), which does a simultaneous alignment of biological sequences with hierarchical clustering. To examine similarity between the sequences, we used Sequence Identity And Similarity, (http://imed.med.ucm.es/Tools/sias.html) with default BLOSUM62 scoring matrices. Evolutionary tree was constructed from the sampled species through Phylogeny.fr (http://www.phylogeny.fr/index.cgi) online program.

### Comparative physicochemical properties of amino acid sequence in the CD14 molecule

The biochemical properties of the amino acids from the 14 mammalian species were computed with ProtParam (www.expasy.org/protparam/). The following properties were computed for each sequence: aliphatic index, which defines the relative volume of a protein occupied by alanine, valine, isoleucine, and leucine; instability index, which estimates the protein stability based on the amino acid composition; protein net charge, which can be positive, negative or neutral based on the amino acid composition in the protein; molecular weight; grand average of hydropathicity (GRAVY), which determines the hydrophobicity of a protein from the aliphatic side chain; and isoelectric point (pI), which is the pH at which the protein net charge is equal to zero.

### Functional analysis, motif scanning and prediction of signal peptides

We performed functional analysis on the protein sequences in order to classify them in to super families, predict domains, repeats and find important sites that may be relevant in evolution. We scanned for the motif signatures among the amino acid sequences with the combined use of ScanProsite (https://prosite.expasy.org/) (Sigrist et al., 2010) and InterPro, an online program that analyzes protein sequences and classification (https://www.ebi.ac.uk/interpro/). The HAMAP profiles, PROSITE patterns, Pfam global models and PROSITE profiles were all included in the search. Sequence logo of the identified conserved domain in the CD14 protein among the 14 mammalian species was constructed with WebLogo (http://weblogo.berkeley.edu/logo.cgi), to show the graphical view of the region containing the conserved amino acid among the species. Furthermore, we predicted the cleavage sites and the presence of signal peptides in CD14 protein from the 14 mammalian species using SignalP 5.0 server (http://www.cbs.dtu.dk/services/SignalP/), which uses recurrent neural network architecture and deep convolution to classify signal peptides into lipoprotein signal peptides, secretory signal peptides or Tat signal peptides (Käll, Krogh & Sonnhammer, 2004). In order to gain a better understanding of the

localization of the protein in each species, we predicted subcellular localizations of CD14 protein using Neural Networks algorithm on DeepLoc-1.0 server (http://www.cbs.dtu.dk/services/DeepLoc/), and the construction of the subcellular pathway hierarchical tree.

## Prediction analysis of amino acid substitution

The effect of the amino acid substitution was predicted using the combination of sorting intolerance from tolerance (SIFT), protein analysis through evolutionary relationship (PANTHER) and protein variation effect analyzer (PROVEAN). Briefly, we used the human CD14 amino acid sequence to query the MSA of other mammalian species in this study using SIFT which predict the tolerance or deleterious effect of substitutions for each position in the query sequence. Any position with probability less than 0.05 is classified as deleterious, as previously described (*Bendl et al., 2014*; *Choi & Chan, 2015*). We selected a total of 10 variants from the mutational hotspots as predicted by SIFT and further estimate the likelihood of the selected variants and their effects on protein function through PROVEAN and PANTHER.

## Prediction of protein interactome with CD14 protein in different species

In order to establish specific interaction of the CD14 protein with other molecules as a result of biochemical events during speciation, we used the retrieved CD14 amino acid sequence from each mammalian species in this study to predict its association with other protein groups and generate different networks using STRING, a database that predicts protein–protein interactions (PPI) (https://string-db.org/). This is important in order to examine the diversity shaped by evolution in the association of the CD14 gene with other molecules in different organisms. Venn diagrams were constructed for the comparison and visualization of overlapping PPI among different species using two web-based applications (http://bioinformatics.psb.ugent.be/software/details/Venn-Diagrams and http://bioinfogp.cnb.csic.es/tools/venny/).

# RESULTS

## Comparative analysis and sequence evolutionary trace

In this study, we examined the evolutionary pattern of CD14 protein sequences in 14 mammalian species. The alignment is conserved within two groups separated into ruminants and non-ruminants. The MSA identified leucine (L), aspartic acid (D), lysine (K), glutamic acid (E), valine (V), glycine (G), serine (S), and asparagine (N) as evolutionarily conserved amino acid residues, while others like proline (P), glutamine (Q), methionine (M), alanine (A), phenylalanine (F), isoleucine (I), and threonine (T) were evolutionarily varied. The CD14 protein sequence demonstrates significant variability in both percentage identity and similarity across the 14 species, despite the common evolutionary origin (Figs 1 and 2). The percentage identity of CD14 protein in monkey, gorilla, chimpanzee, and human was similar while gorilla shares the closest identity with human (Table 1). Among the ruminants, cattle, and yak share the closet similarity compared to buffalo, sheep and goat, although the phylogenetic tree suggests that goat is

Figure 1 — Multiple sequence alignment of CD14 promoter regions between mammalian species.

```
            1                                                                                                                                    130
HUMAN       MERASCLLLL LL?-LVHVSA TTPEPCELDD ED--FRCVCN FSEPQPDWSE AFQCVSAVEV EIHAGGLNLE PFLKRVDADA DPRQYADTVK ALRVRRLTVG AAQVPAQLLV GALRVLAYSR LKELTLEDLK
GORILLA     MGRASCLLLL LL?-LVHVSA TTPEPCELDD ED--FRCVCN FSEPQPDWSE AFQCVSAVEV EIRAGGLNLE PFLKRVDADA DPRQYADTVK ALRVRRLTVG AAQVPAQLLV GALRVLAYSR LKELTLEDLK
CHIMPANZEE  MERASCLLLL LL?-LVHVSA TTPEPCELDD ED--FRCVCN FSEPQPDWSE AFQCVSAVEV EIRAGGLNLE PFLKRVDADA DLRQYADTVK ALRVRRLTVG AAQVPAQLLV VALRVLAYSR LKELTLEDLK
MONKEY      MERASCLLLL LL?-LVHVSA TTPEPCELDD ED--FRCVCN FSEPHPDWSE AFQCVSAVEV EIRVGGLSLE PFLTRVDPDA DPRQYADTIK ALRVRRLTVG AAQVPAQLLV GALRVLAYSR LQELTLEDLE
RABBIT      MEPVPCLLLL LL?-LLRAST DTPEPCELDD DD--IRCVCN FSDPQPDWSS ALQCMPAVQV EMWGGGHSLE QFLRQADLYT DQRRYADVVK ALRVRRLTVG AVQVPAPLLL GVLRVLGYSR LKELALEDIE
HORSE             LLL LL?-LLRFSA ATLEPCEVDD EN--FRCVCN FTGPQPDWSS AFQCMTAVEV EIRGGGRNLE QFLKGAS--A DPKQYADIVK ALRLQRLTVG AVQVPAPLLV ALLRALGYSR LKELTLEDLE
BOVINE      MVCVPYLLLL LL?SLLRVSA DTTEPCELDD DD--FRCVCN FTDPKPDWSS AVQCMVAVEV EISAGGRSLE QFL-K-GADT NPKQYADTIK ALRVRRLKLG AAQVPAQLLV AVLRALGYSR LKELTLEDLE
YAK         QVCVPYLLLL LL?SLLRVSA DTTEPCELDD DD--FRCVCN FTDPKPDWSS AVQCMVAVEV EISAGGRSLE QFL-K-GADT NPKQYADTIK ALRVRRLKLG AAQVPAQLLV AVLRALGYSR LKELTLEDLE
BUFFALO     MVCVPYLLLL LL?PLLRVSA DTTEPCELDD DD--FRCVCN FTDPKPDWSS AVQCMVAVEV EISGGGRSLE QFL-K-GADT NPKQYADTIK ALRVRRLKLG AAQVPAQLLV AVLRALGYSR LKELTLEDLE
SHEEP        VCVPSLLLL LLPPLLRVSA DTTEPCELDD DD--FRCVCN FTDPKPDWSS AVQCMVAVEV EIRGGGHSLD QFL-K-GVNT DPKQYADTIK ALRVRRLKLG AAQVPAQLLV AVLRALGYSR LKELTLEDLE
GOAT        MVCVPYLLLL LLSALLRVTA DKREPCELDP QH--FRCVCN FTDPKPDWSS AVQCMVAVEV EIRGGGHSLD QFL-K-GANT DPKQYADTIK ALRVRRLKLG AAQVPAQLLV AVLRALGYSR LKELTLEDLE
PIG         MVRLPCPLLL LL?-VRCVCN ATPEPCQIDD ED--VRCVCN FTHPQPDWSS ALQCVAAVEV EIRGGGHSLD EFLLK-SASA NPKQYADMLK ALRLRRLTVG AARVPAQILA LVLHALGFSR LKELTLEDLE
RAT         MKLMLGLLL LPLTLVHASP ATEPCELDQ DEESVRCYCN FSDPQPNWSS AFLCAGAEDV EFYGGGRSLE YLLKRVDTEA NLGQYTDIIR SLPLKRLTVR SARVPTQILF GTLRVLGYSG LRELTLENLE
MOUSE       MERVLGLLL L--LLVHASP APPEPCELDE ES----CSCN FSDPKPDWSS AFNCLGAADV ELYGGGRSLE YLLKRVDTEA DLGQFTDIIK SLSLKRLTVR AARIPSRILF GALRVLGISG LQELTLENLE
Consensus   m.....1LLL Llp.LlrvSa .tpEPC#1D# #d..frCvCN FsdPqP#WSs AfqC..Av#V Ei.gGGrsL# .fL...d..a #p.q%aD.ik aLrvrRLtvg aaq!PaqlLv ..LrvLgySr LkELtLE#le

            131                                                                                                                                  260
HUMAN       ITGTMPPLP- LEA------- -TGLALSSLR LRNVSWATGR SWLAELQQWL KPGLKVLSIA QAHSPAFSCE QVRAFPALTS LDLSDNPGLG ERGLMAALCP HKFPAIQNLA LRNTGMETPT GVCAALAAAG
GORILLA     ITGTMPPLP- LEA------- -TGLALSSLR LRNVSWATGR SWLAELQQWL KPGLKVLSIA QAHSPAFSCE QVRAFPALTS LDLSDNPGLG ERGLIAALCP HKFPAIQNLA LRNTGMETPT GVCAALAAAG
CHIMPANZEE  ITGTMPPLP- LEA------- -TGLALSSLR LRNVSWATGR SWLAELQQWL KPGLKVLSIA QAHSPAFSCE QVRAFPALTS LDLSDNPGLG ERGLIAALCP HKFPAIQNLA LRNTGMETPT GVCAALAAAG
MONKEY      ITGTMPPLP- LEA------- -TGLALSSLR LHNVSWATGR SWLAELQQWL KPGLKVLSIA QAHSPAFSCE QVRAFPALTS LDLSDNPGLG ERGLIAALCP HKFPALQNLA LRNTGMETPT GVCAALAAAG
RABBIT      VTGTAPPPPP LEA------- -TGPALSTLS LRNVSWPKGG AWLSELQQWL KPGLQVLNIA QAHTLAFSCE QVRTFSALTT LDLSENPGLG ERGLVAALCP HKFPALQDLA LRNAGMKTLQ GVCAALAEAG
HORSE       VTGTMQPPP- LEA------- -TGPPLSSLR LRNVSWATGG AWLAELQQWL KPGLKILSIA QAHSLAFSCE QLHSFSALHT LDLSDNPGLG ERGLIAALCP HKFPALRDLA LRNAGMQTPN GVCAAMAAAG
BOVINE      VTGTPTPPTP LEA------- -AGPALTTLS LRNVSWTTGG AWLGELQQWL KPGLRVLNIA QAHSLAFPCA GLSTFEALTT LDLSDNPSLG DSGLMAALCP NKFPALQYLA LRNAGMETPS GVCAALAAAR
YAK         VTGTPTPPTP LEATPPPLLK AAGPALTTLS LRNVSWTTGG AWLGELQQWL KPGLRVLNIA QAHSLAFPCA GLSTFEALTT LDLSDNPSLG DSGLMAALCP NKFPALQYLA LRNAGMETPS GVCAALAAAR
BUFFALO     VTGPMPPKP- LEA------- -TGPAFTTLS LRNVSWATGG AWLGELQQWL KPGLRVLNIA QAHSLAFPCA GLSTFEALTT LDLSDNPSLG DTGLMAALCP NKFPALQCLA LRNAGMDKLS GVCAALAAAR
SHEEP       VTGTPTPPTP LEA------- -TGPALTTLS LRNVSWATGG AWLGELQQWL KPGLRALNIA QAHSLAFPCA GLSTFEALTT LDLSDNPSLG DSGLMAALCP NKFPALQYLA LRNAGMETPS GVCAALAAAR
GOAT        VTGTPTPPAP LEA------- -TGPALTTLS LRNVSGTTGG AWLGELQPWL KPGLRALNIA QAHSLAFPCA GLSTFEALTT LDLSDNPSLG DSGLMAALCP NKFRAPQYLA LRNAGMEAAT RPCAPLAAAR
PIG         VTGQVPPPL- QET------- -PGPALTTLR LRNVSWATGG AWLGELQQWL QPSLKVLKVA QASSLAFPCA QLRAFPALTT LDLSDNPGLG ERGLTAALCP RKFPALEDLA LRNAGVETPS GVCAALAGAG
RAT         VTGTALSPL- LDA------- -TGPDLNTLS LRNVSWATTD TWLAELQQWL KPGLKVLSIA QAHSLNFSCK QVGVFPALAT LDLSDNPELG EKGLISALCP HKFPTLQVLA LRNAGMETTS GVCSALAAAR
MOUSE       VTGTAPPPL- LEA------- -TGPDLNILN LRNVSWATRD AWLAELQQWL KPGLKVLSIA QAHSLNFSCE QVRVFPALST LDLSDNPELG ERGLISALCP LKFPTLQVLA LRNAGMETPS GVCSALAAAR
Consensus   !TGt.pppp. l#a         tGpaL.tL. LRNVSWatgg aWLaELQQWL kPgLkvLs!A QAhslaFsC. qvr.FpALtt LDLS#NPgLG #rGL.aALCP hKFPalq.LA LRNaGmetps GVCaA$AaAg

            261                                                                                                                           387
HUMAN       VQPHSLDLSH NSLRATVNPS APRCMWSSAL NSLNLSFAGL EQVPKGLPAK LRVLDLSCNR LNRAPQPDEL PEVDNLTLDG NPFLVPGTAL PHEGSMNSGV VPACARSTLS VGVSGTLVLL QGARGFA
GORILLA     VQPHSLDLSH NSLRATVNPS APRCMWSSAL NSLNLSFAGL EQVPKGLPAK LRVLDLSCNR LNRAPQPDEL PEVDNLTLDG NPFLVPGTAL PHEGSMNSGV VPACARSTLS VGVSGTLVLL QGARGFA
CHIMPANZEE  VQPHSLDLSH NSLRATVNPS APRCMWSSAL NSLNLSFAGL EQVPKGLPAK LRVLDLSCNR LNRAPQPDEL PEVDNLTLDG NPFLVPGTAL PHEGSMNSGV VPACARSTLS VGVSGTLVLL QGARGFA
MONKEY      VQPHSLDLSH NSLRATANPS APRCMWSSAL NSLNLSFAGL EQVPKGLPAK LRVLDLSCNR LNRRPRPDEL PQVDNLALDG NPFLVPGTAL PQEGSMNSGV VPACARSTLS VGVSGTLVLL QGARGFA
RABBIT      VQPHHLDLSH NSLRADTQ-- --RCIWPSAL NSLNLSFTGL QQVPKGLPAK LNVLDLSCNK LNRAPQPDEL PKVVNLSLDG NPFLVPGASK LQEDLTNSGV FPACPPSFLA MGMSGTLALL QGARGFI
HORSE       VQPHSLDLSH NSLSAAA-PG APRCDWPSAL SSLNLSFAGL EQVPKGLPGK LSVLDLSCNR LNKAPRADEL PVVSNLILDR NPYLDPEASK QQD--QNSGV VAACAHSALT VGISGTLALL RGAGDFA
BOVINE      VQPQSLDLSH NSLRVTA-PG ATRCVWPSAL RSLNLSFAGL EQVPKGLPPK LSVLDLSCNK LSREPRRDEL PEVNDLTLDG NPFLDPGALQ HQNDPMISGV VPACARSALT MGVSGALALL QGARGFA
YAK         VQPQSLDLSH NSLRVTA-PG ATRCVWPSAL RSLNLSFAGL EQVPKGLPPK LSVLDLSCNK LSREPRRDEL PEVNDLTLDG NPFLDPGALQ HQNDPMISGV VPACARSALT MGVSGALALL QGARGFA
BUFFALO     VQPQSLDLSH NSLRVTA-PG ATRCVWPSAL RSLNLSFAGL EQVPKGLPPK LSVLDLSCNK LSREPRRDEL PEVNDLTLDG NPFLDPGALQ RQNDPMISGV VPACARSALT MGVSGALALL QGARGFA
SHEEP       VQPQSLDLSH NSLRVT--PG ATRCVWPSAL SSLNLSFAGL EQVPKGLPTK LSVLDLSCNK LSREPRREL PEVNVLTLDG NPFLDPGALK HQDDPMISGV VPACARSALT MGVSGALALL QGARGFA
GOAT        VQPQNLDLSH NSLRVTA-PG ATRCVWPSAI SSLNCSFAGL EQVPKGLPPK LSVLDLSCNK LSREPRRDEL PYVNVLTVNG NPFLDPGALQ HQNDPMISRM IPDRAPPSLV IGVSGALVLY QGARGFA
PIG         VQPHRLDLSH NSLRATA-AG ARECVWPAAL SSLNLSFAEL EQVPKGLPPK LTVLDLSCNK LNRAPRPEEL PAVDDLTLEG NPYMDPEALQ HQEDPMASGV VPPCARSALT MGVSGTLALL QGARGFA
RAT         VPLQALDLSH NSLRDTA--G TPSCDWPSQL NSLNLSFTGL EHVPKGLPAK LSVLDLSYNR LDRKPRPEEL PEVGSLSLTG NPFLHSES-- -QSEAYNSGV VIATALSPGS AGLSGTLALL LGHRLFV
MOUSE       VQLQGLDLSH NSLRDAA--G APSCDWPSQL NSLNLSFTGL KQVPKGLPAK LSVLDLSYNR LDRNPSPDEL PQVGNLSLKG NPFLDSES-- -HSEKFNSGV VTAGAPSSQA VALSGTLALL LGDRLFV
Consensus   Vqph.LDLSH NSLrata.pg aprC.WpsaL nSLNLSFagL eqVPKGLPaK LsVLDLScNr Lnr.PrpdEL P.V.nL.Ldg NP%$dpga.. .q...mnSGV vpaca.S.l. .gvSGtLaLL qGargFa
```

**Figure 1 Multiple sequence alignment of CD14 promoter regions between mammalian species.**

distantly related. While mouse and rat cluster with the same origin, the analysis show that they share less identity (7.4%) and similarity (13.4%). Rabbit, horse and pig are distantly apart from other species, as they do not share high conservation (Table 1; Fig. 2). In all, the sequence of CD14 protein in goat and horse share the least identity (6.7% and 6.9% for goat and horse, respectively) and similarity (9.9% and 13.2% for goat and horse respectively) with human.

## Physicochemical properties at the CD14 promoter region

The ProtParam tool (www.expasy.org/protparam/) was used to compute the physical and chemical properties of CD14 amino acid sequences among the 14 species (Table 2). The aliphatic index of all the species is generally high for all species showing that the protein is thermally stable. A higher instability index was observed in the CD14 molecule
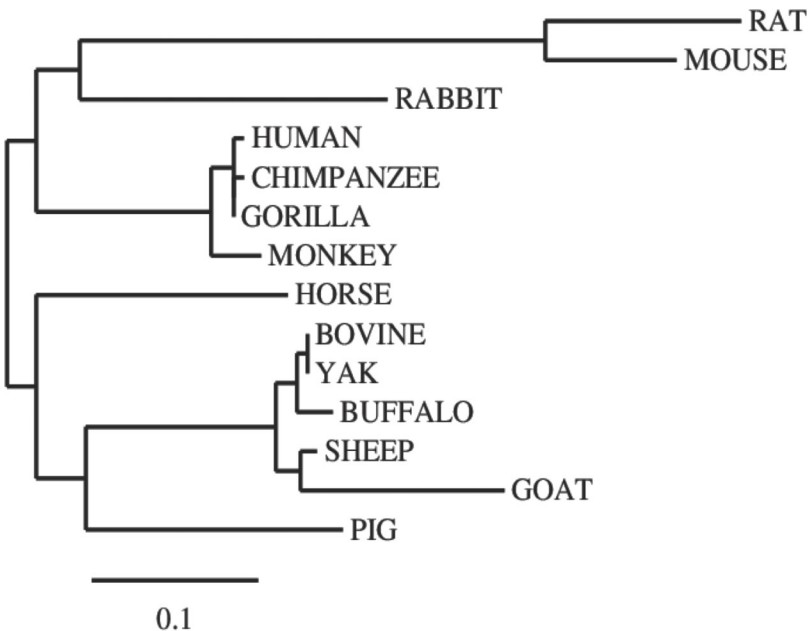

**Figure 2 Phylogenetic tree of evolutionary relationships among taxa.** The evolutionary history was inferred using the Neighbor-Joining method. The optimal tree with the sum of branch length = 1.48602764. The tree is drawn to scale, with branch lengths in the same units as those of the evolutionary distances used to infer the phylogenetic tree. The evolutionary distances were computed using the *p*-distance method and are in the units of the number of amino acid differences per site. The analysis involved 14 amino acid sequences. The coding data were translated assuming a standard genetic code table.

of rabbit, pig, and monkey (53.0, 46.8, and 45.1, respectively), indicating that the protein is less stable and hydrophobic amino acids such as leucine, valine, serine, and asparagine, occupy majority of the sequence, providing higher tolerance against diseases. The lowest instability index is observed in horse (33.5) and goat (35.1) showing that the protein is more stable in these species. The CD14 protein in goat also has the lowest aliphatic index (99.7) while mouse has the highest (107.7). We observed a closer range of molecular weight among the species in this study, although gorilla, monkey, human, chimpanzee, and rat had the higher molecular weight with close range (Table 2). Negative net charge, indicative that the protein is more basic than acidic, ranged from −9 to as found in mouse and rat to +4 as found in goat. Goat, horse and gorilla has higher isoelectric point indicating that the CD14 molecule is highly basic in these species than others. The GRAVY values obtained were generally positive and higher in ruminants than non-ruminants suggesting the proteins are more hydrophobic, which enhances oligomerization and higher binding capability to different proteins.

## Characterization of functional motifs and prediction of signal peptides

The CD14 amino acid sequences of the 14 mammalian species in this study were individually scanned for matches against the InterPro and PROSITE collection of protein signature databases. We found one domain (Leucine-rich repeat (LRR), PS51450) with varying frequency across the 14 species (Fig. 3). Comparison of the predicted intra-domain

**Table 1 Percentage identity (similarity) of the CD14 protein across mammalian species.**

| Human | 100 (100) | | | | | | | | | | | | | |
|---|---|---|---|---|---|---|---|---|---|---|---|---|---|---|
| Rat | 23.1(27.9) | 100 (100) | | | | | | | | | | | | |
| Mouse | 10.4 (14.5) | 7.4 (13.4) | 100 (100) | | | | | | | | | | | |
| Cattle | 8.3 (11.5) | 10.2 (15.1) | 10.9 (15.3) | 100 (100) | | | | | | | | | | |
| Rabbit | 29.3 (33.6) | 15.6 (21.5) | 9.3 (15.0) | 9.4 (13.4) | 100 (100) | | | | | | | | | |
| Goat | 6.7 (9.9) | 9.9 (15.1) | 9.7 (13.7) | 87.4 (89.3) | 10.2 (14.2) | 100 (100) | | | | | | | | |
| Monkey | 95.2 (96.3) | 23.1 (28.0) | 10.7 (14.5) | 8.6 (12.1) | 29.0 (33.9) | 6.9 (10.5) | 100 (100) | | | | | | | |
| Gorilla | 99.2 (99.5) | 23.1 (28.0) | 10.4 (14.2) | 8.3 (11.5) | 29.0 (33.6) | 6.7 (9.9) | 95.5 (96.3) | 100 (100) | | | | | | |
| Sheep | 20.8 (26.4) | 12.4 (17.0) | 7.9 (13.9) | 8.9 (10.5) | 19.4 (25.3) | 8.6 (10.8) | 21.3 (27.0) | 21.0 (26.4) | 100 (100) | | | | | |
| Horse | 6.9 (13.2) | 11.3 (17.1) | 7.7 (13.8) | 8.8 (13.8) | 8.3 (14.0) | 8.8 (13.5) | 6.6 (12.9) | 6.9 (11.8) | 6.9 (11.8) | 100 (100) | | | | |
| Pig | 18.5 (23.3) | 13.2 (19.1) | 10.9 (14.5) | 67.6 (71.6) | 18.8 (23.1) | 60.1 (64.9) | 19.3 (23.6) | 18.8 (23.3) | 19.1 (22.9) | 8.0 (13.2) | 100 (100) | | | |
| Buffalo | 8.0 (11.3) | 9.9 (14.8) | 10.7 (15.3) | 96.5 (97.3) | 9.4 (13.2) | 86.1 (88.7) | 8.3 (11.8) | 8.0 (11.3) | 8.9 (10.8) | 8.5 (12.9) | 66.8 (71.3) | 100 (100) | | |
| Chimp | 98.9 (99.2) | 23.1 (28.0) | 10.9 (14.8) | 8.6 (11.8) | 29.0 (33.6) | 6.9 (10.2) | 95.2 (96.0) | 99.2 (99.2) | 21.0 (26.7) | 6.9 (13.2) | 19.0 (23.6) | 8.3 (11.5) | 100 (100) | |
| Yak | 8.3 (12.0) | 8.6 (14.2) | 9.0 (12.3) | 42.1 (45.3) | 8.3 (14.0) | 37.5 (42.1) | 8.3 (12.3) | 8.3 (12.0) | 9.4 (13.2) | 8.0 (14.0) | 21.4 (26.5) | 41.0 (44.5) | 8.5 (12.3) | 100 (100) |
| | **Human** | **Rat** | **Mouse** | **Cattle** | **Rabbit** | **Goat** | **Monkey** | **Gorilla** | **Sheep** | **Horse** | **Pig** | **Buffalo** | **Chimp** | **Yak** |

Notes:
Identity: Minimum = 6.61; Maximum = 99.2; Mean = 23.26; Standard deviation = 26.56.
Similarity: Minimum = 9.91; Maximum = 100; Mean = 32.58; Standard deviation = 30.99.

**Table 2 Physicochemical properties of the CD14 promoter region in selected mammalian species.**

| Species | Amino acids size | Molecular weight (Da) | Isoelectric point | Instability index | Aliphatic index | Net charge | GRAVY |
|---|---|---|---|---|---|---|---|
| Chimpanzee | 375 | 40,135.34 | 5.92 | 43.44 | 104.61 | −4 | 0.113 |
| Gorilla | 375 | 40,005.15 | 6.10 | 42.27 | 102.80 | −3 | 0.094 |
| Human | 375 | 40,076.20 | 5.84 | 42.93 | 101.76 | −5 | 0.083 |
| Monkey | 375 | 40,127.19 | 5.69 | 45.10 | 102.80 | −6 | 0.085 |
| Horse | 363 | 38,450.27 | 6.19 | 33.47 | 103.06 | −3 | 0.096 |
| Mouse | 366 | 39,203.94 | 5.08 | 41.16 | 107.70 | −9 | 0.051 |
| Pig | 373 | 39,724.01 | 5.82 | 46.83 | 103.40 | −4 | 0.073 |
| Rabbit | 372 | 39,992.29 | 5.72 | 52.99 | 103.33 | −5 | 0.041 |
| Rat | 372 | 40,053.85 | 5.33 | 40.19 | 104.11 | −9 | 0.033 |
| Buffalo | 373 | 39,756.09 | 5.84 | 41.49 | 101.80 | −2 | 0.099 |
| Cattle | 373 | 39,666.79 | 5.37 | 41.70 | 102.06 | −5 | 0.099 |
| Goat | 373 | 39,930.28 | 8.47 | 35.07 | 99.71 | +4 | 0.032 |
| Sheep | 371 | 39,368.43 | 5.50 | 40.27 | 101.54 | −5 | 0.087 |
| Yak | 381 | 40,481.75 | 5.54 | 41.63 | 102.23 | −4 | 0.082 |

features show one LRR domain in human, two each in gorilla, chimpanzee, monkey, horse and pig, three each in cattle, sheep, buffalo, yak, and mouse, with the highest number (4) found in rat. Figure 4 shows the MSA of the homology of LRR domain across the 14 species, showing that leucine, aspartic acid, serine, and asparagine are 100% conserved in this region. The sequence logo built from the MSA of the domain is displayed in Fig. 5, with the logo showing the relative frequencies of each conserved amino acid and their position in the LRR domain. The domain homology reveals that there is significant conservation of most amino acids in this region.

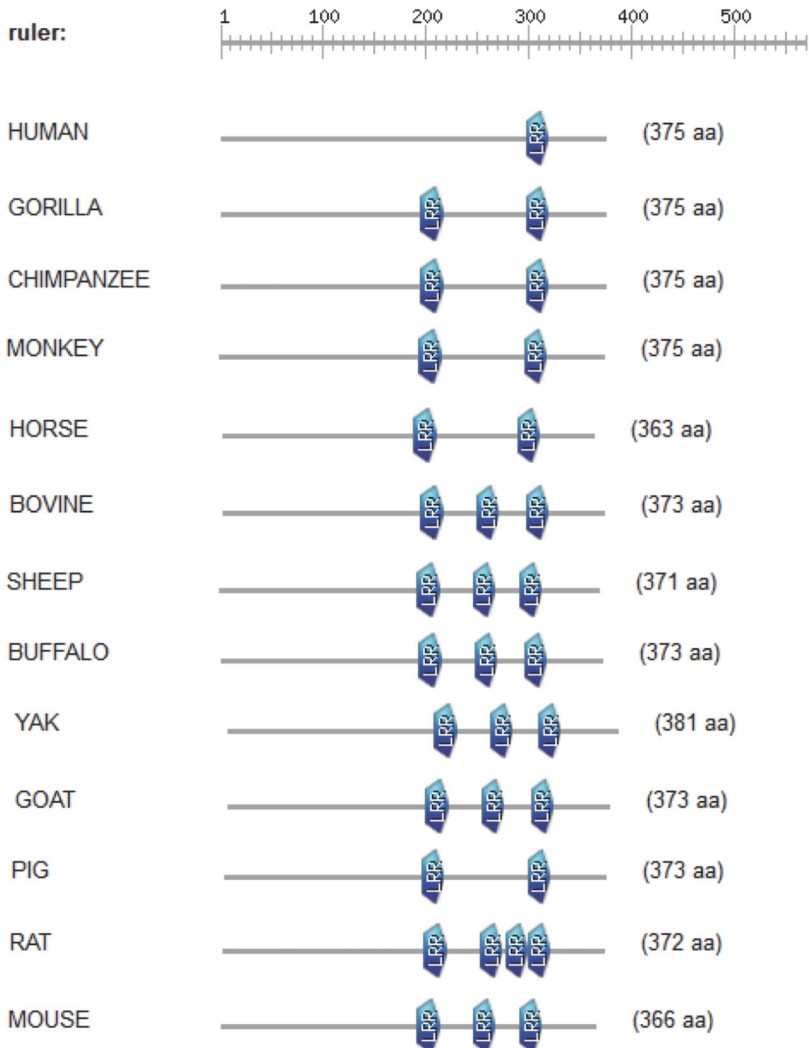

**Figure 3 Comparison of predicted intra-domain features of CD14 protein.** This comparison show leucine-rich repeat (PS51450), which provides additional information about the structure and function of critical amino acids in the 14 mammalian species.

Furthermore, we predicted the signal peptides, position, and secretory pathway of the CD14 amino acids in the 14 species under consideration. Our analysis shows that chimpanzee, gorilla, human, and monkey share the same signal peptide (VSA-TT) at the same position (19 and 20), with high likelihood (Table 3). Buffalo, cattle, sheep, and yak also share the same signal peptide (VSA-DT) and position (20 and 21) although sheep has a different position (19 and 20). We observed a significant variation for the rest of the species in terms of signal peptides and their positions (Table 3). Interestingly, signal peptide for all the species (Fig. 6A), except sheep (Fig. 6B), share the same subcellular localization in the neural networks.

## Mutational analysis of predicted variation

A total of 10 variants were selected from the predicted mutations by SIFT and the effects were tested as deleterious or not in the 14 species with PROVEAN and PANTHER.
```
HUMAN         KLRVLDLSCN.R.LNRAPQPDELP
RAT           ALATLDLSDNpE.LGERGLISALC
RAT_2         PLQALDLSHN.S.LRDTAGTPSCD
RAT_3         KLSVLDLSYN.R.LDRKPRPEELP
RAT_4         QLNSLNLSFT.G.LEHVPKGLPA-
MOUSE         ALSTLDLSDNpE.LGERGLISALC
MOUSE_2       QLQGLDLSHN.S.LRDAAGAPSCD
MOUSE_3       KLSVLDLSYN.R.LDRNPSPDELP
BOVINE        ALTTLDLSDN.PsLGDSGLMAALC
BOVINE_2      QPQSLDLSHN.S.LRVTAPGATRC
BOVINE_3      KLSVLDLSCN.K.LSREPRRDELP
GOAT          ALTTLDLSDN.PsLGDSGLMAALC
GOAT_2        QPQNLDLSHN.S.LRVTAPGATRC
GOAT_3        KLSVLDLSCN.K.LSREPRRDELP
MONKEY        ALTSLDLSDN.PgLGERGLIAALC
MONKEY_2      KLRVLDLSCN.R.LNRRPRPDELP
GORILLA       ALTSLDLSDN.PgLGERGLIAALC
GORILLA_2     KLRVLDLSCN.R.LNRAPQPDELP
SHEEP         ALTTLDLSDN.PsLGDSGLMAALC
SHEEP_2       QPQSLDLSHN.S.LRVTPGATRCV
SHEEP_3       KLSVLDLSCN.K.LSREPRREELP
PIG           ALTTLDLSDN.P.GLGERGLTAAL
PIG_2         KLTVLDLSCN.K.LNRAPRPEELP
BUFFALO       ALTTLDLSDN.PsLGDTGLMAALC
BUFFALO_2     QPQSLDLSHN.S.LRVTAPGATRC
BUFFALO_3     KLSVLDLSCN.K.LSREPRRDELP

CHIMPANZEE    ALTSLDLSDNPgLGERGLIAALC
CHIMPANZEE_2  KLRVLDLSCNR.LNRAPQPDELP
YAK           ALTTLDLSDNPsLGDSGLMAALC
YAK_2         QPQSLDLSHNS.LRVTAPGATRC
YAK_3         KLSVLDLSCNK.LSREPRRDELP
HORSE         ALHTLDLSDNPgLGERGLIAALC
HORSE_2       KLSVLDLSCNR.LNKAPRADELP
```

**Figure 4 Conserved domain LRR patterns across mammalian species.**

Our analysis showed that four of these variants (D28V, W45H, G62E, L70D) were validated mutations with deleterious effect on all species with two others found in few species. These variants cluster in the C-terminus region of CD14 protein between 20 and 100 amino acids. A closer look suggests that mutational effect on the CD14 protein sequence varied from C-terminus to N-terminus with less mutational effect toward the N-terminus (Table 4). The deleterious mutations observed in our study were all at the C-terminus region thus identifying it as a mutational hotspot. Q100G, V301M, L318I, G335T, L357H, and G370K mutation spots were neutral for most species. This might mean that CD14 is less conserved in this region because of evolutionary divergence of all species.

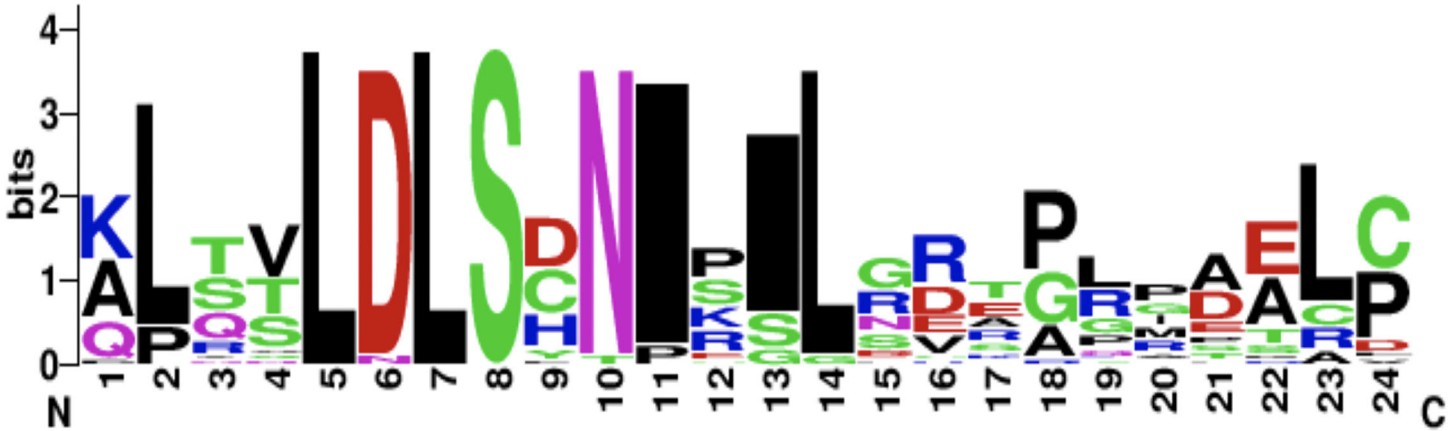

**Figure 5 CD14 protein sequence logo displaying the most conserved domain and the positions of amino acids.** Sequence logo displaying the most conserved domain and the positions of amino acids starting from the N-terminus on the left to C-terminus to the right. The relative frequency of the amino acids is shown on the *y*-axis.               

**Table 3 Prediction of signal peptides and properties of the CD14 molecule in selected mammalian species.**

| Species | Amino acids size | Cleavage position | Signal site | Probability | Likelihood | Others |
|---|---|---|---|---|---|---|
| Chimpanzee | 375 | 19 and 20 | VSA-TT | 0.9140 | 0.9991 | 0.0009 |
| Gorilla | 375 | 19 and 20 | VSA-TT | 0.9077 | 0.9991 | 0.0009 |
| Human | 375 | 19 and 20 | VSA-TT | 0.9142 | 0.9991 | 0.0009 |
| Monkey | 375 | 19 and 20 | VSA-TT | 0.9142 | 0.9991 | 0.0009 |
| Horse | 363 | 14 and 15 | AAT-LE | 0.2069 | 0.675 | 0.3250 |
| Mouse | 366 | 17 and 18 | ASP-AP | 0.4563 | 0.9991 | 0.0009 |
| Pig | 373 | 19 and 20 | VSA-AT | 0.7699 | 0.9989 | 0.0011 |
| Rabbit | 372 | 19 and 20 | AST-DT | 0.6574 | 0.9981 | 0.0019 |
| Rat | 372 | 17 and 18 | VHA-SP | 0.8795 | 0.9998 | 0.0002 |
| Buffalo | 373 | 20 and 21 | VSA-DT | 0.9712 | 0.999 | 0.0010 |
| Cattle | 373 | 20 and 21 | VSA-DT | 0.9750 | 0.9992 | 0.0008 |
| Goat | 373 | 20 and 21 | VTA-DK | 0.9642 | 0.9991 | 0.0009 |
| Sheep | 371 | 19 and 20 | VSA-DT | 0.9000 | 0.9453 | 0.0547 |
| Yak | 381 | 20 and 21 | VSA-DT | 0.9752 | 0.9993 | 0.0007 |

However, L-H at position 357 showed a deleterious effect in cattle, yak, pig, gorilla, human, monkey, buffalo, and chimpanzee, while there is also a deleterious effect of G-K at position 370 of CD14 in rat.

## Protein–protein interaction cluster with CD14 gene in different species

In order to deduce PPI that evolved through speciation due to co-localization, additive genetic interaction, co-expression or repression, and physical association with CD14 in the mammalian species under study, we used STRING to build the protein network based on collection of laboratory experimental results from the database (Fig. 7) and segment the
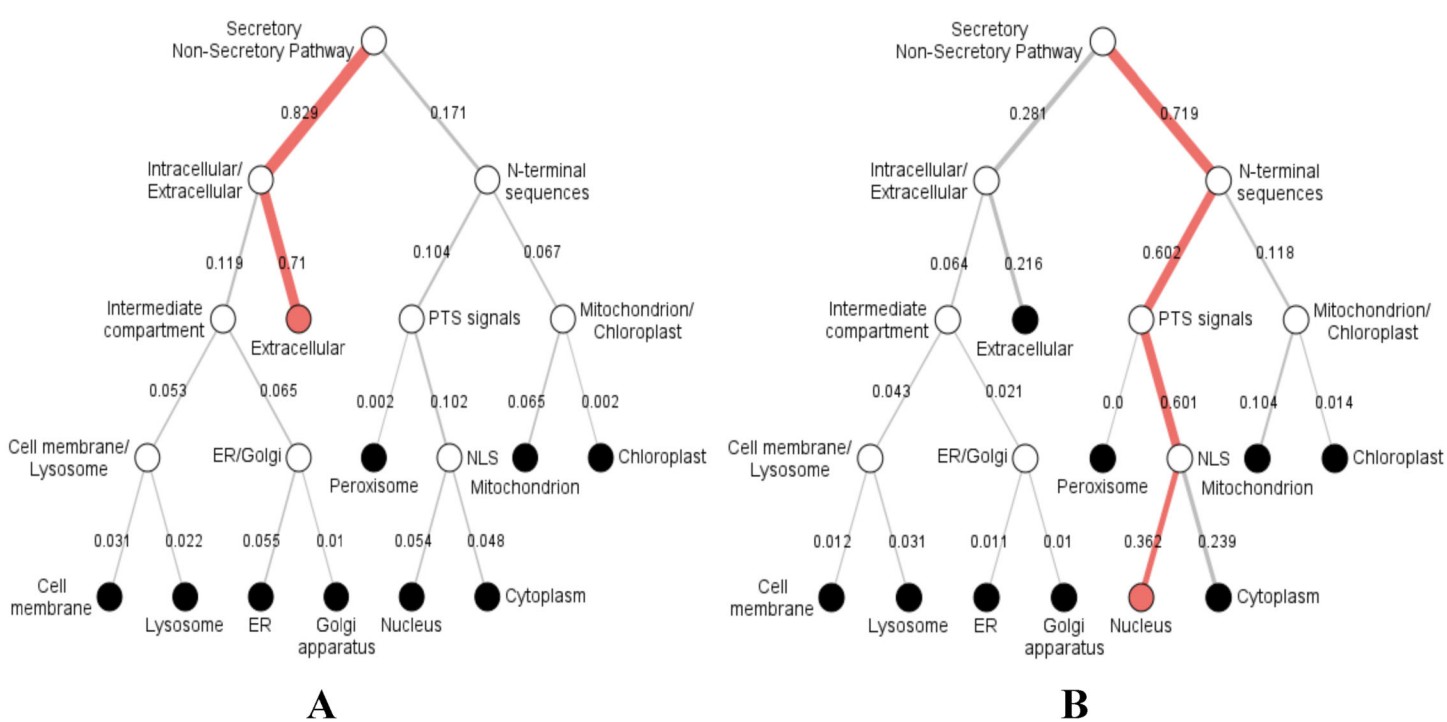

**Figure 6 Hierarchical tree-predicted subcellular localizations of CD14 protein using neural networks algorithm.** (A) Hierarchical tree for all other mammalian species analyzed. (B) Hierarchical tree for sheep only.

**Table 4 Prediction of amino acid mutation at the mutational hotspot of CD14 molecule in selected mammalian species.**

| Species | D28V | W45H | G62E | L70D | Q100G | V301M | L318I | G335T | L357H | G370K |
|---|---|---|---|---|---|---|---|---|---|---|
| Chimpanzee | −3.472 | −4.705 | −3.154 | −3.083 | −2.591 | −1.905 | −1.378 | −1.397 | −3.088 | −2.287 |
| Gorilla | −3.822 | −4.651 | −3.216 | −2.984 | −2.554 | −2.049 | −1.446 | −1.397 | −3.050 | −2.285 |
| Human | −3.679 | −4.680 | −3.008 | −3.056 | −2.756 | −2.043 | −1.445 | −1.395 | −3.229 | −2.305 |
| Monkey | −3.563 | −4.782 | −3.238 | −3.038 | −2.758 | −1.933 | −1.444 | −1.293 | −3.089 | −2.268 |
| Horse | −3.742 | −4.914 | −3.513 | −3.524 | −2.364 | −1.896 | −1.412 | −0.983 | −2.054 | −2.067 |
| Mouse | −3.437 | −4.803 | −3.408 | −1.635 | −2.754 | −2.009 | −1.408 | −1.534 | −2.437 | −1.828 |
| Pig | −3.712 | −5.054 | −3.702 | −1.873 | −2.329 | −2.013 | −1.637 | −1.235 | −2.902 | −2.052 |
| Rabbit | −2.759 | −4.293 | −2.910 | −4.007 | −2.744 | −1.969 | −1.574 | −0.544 | −1.865 | −2.451 |
| Rat | −3.478 | −4.725 | −3.373 | −1.058 | −2.905 | −2.038 | −1.351 | 0.464 | −2.497 | −2.619 |
| Buffalo | −3.310 | −5.083 | −3.497 | −3.130 | −2.169 | −2.064 | −1.390 | −1.427 | −3.065 | −2.213 |
| Cattle | −3.289 | −5.038 | −2.998 | −2.991 | −2.095 | −2.131 | −1.385 | −1.758 | −2.634 | −2.191 |
| Goat | −3.919 | −4.906 | −3.964 | −3.390 | −2.461 | −2.046 | −1.476 | −0.631 | −1.439 | −1.601 |
| Sheep | −3.559 | −4.952 | −4.072 | −3.206 | −2.312 | −1.981 | −1.246 | −1.376 | −2.335 | −1.695 |
| Yak | −3.229 | −5.036 | −3.081 | −3.188 | −2.233 | −2.097 | −1.385 | −1.575 | −2.668 | −2.225 |

Notes:
Prediction (cutoff = −2.5); values above cutoff are considered deleterious; values below cutoff are considered neutral. Red color values indicate deleterious mutations and black color values indicate neutral mutations.

gene pool base on our phylogenetic result to build Venn diagrams for each species cluster (Figs. 8A–8C). We could not find any protein network for horse and so was excluded in the analysis. Our result shows that there is significant variation in the CD14 protein interactome

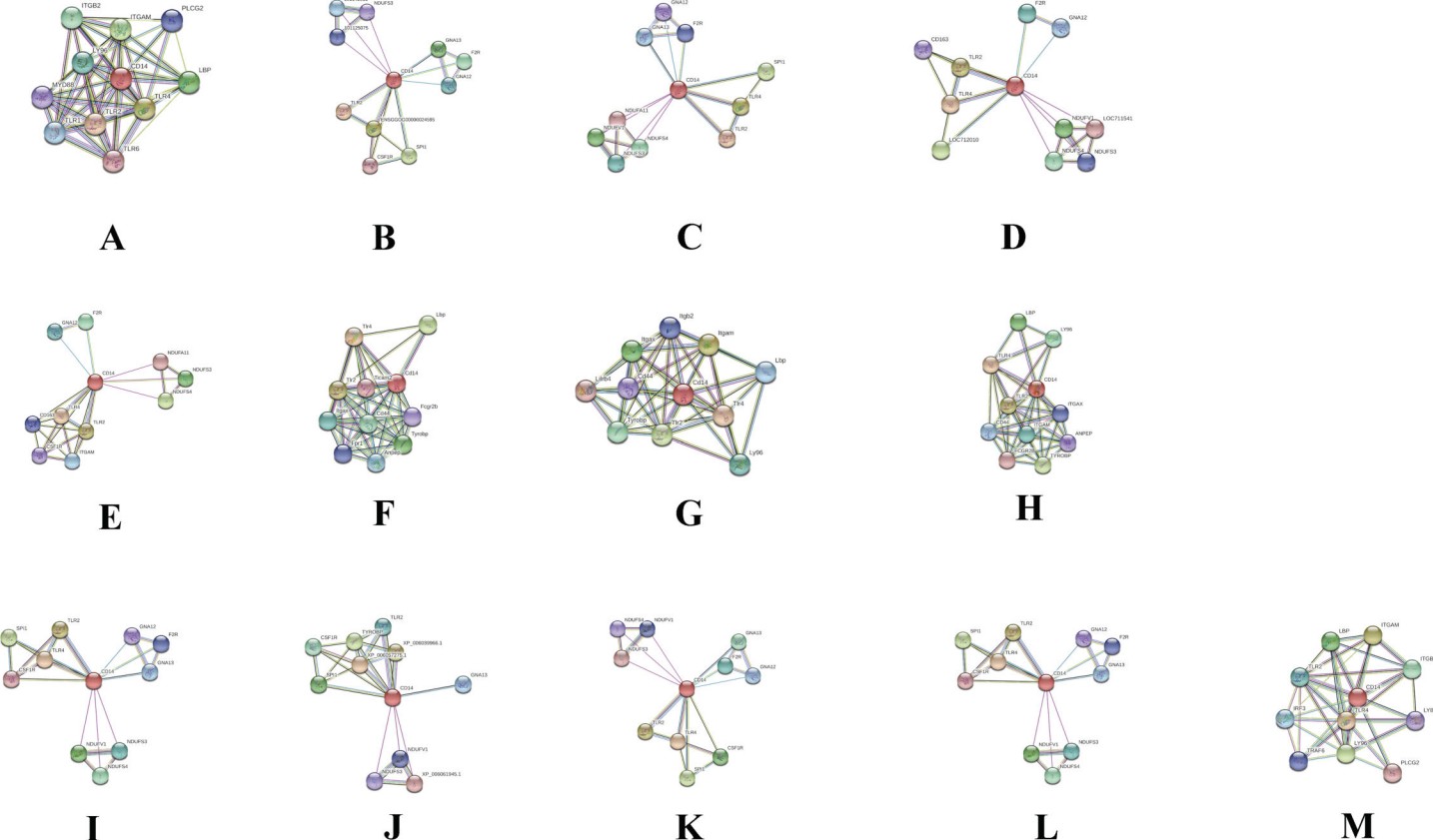

**Figure 7 Network view of predicted associations for group of proteins with CD14.** The network nodes are proteins. The edges represent the predicted functional associations.The thickness of the line indicate the degree of confidence prediction for the interaction. Red line, indicates the presence of fusion evidence; Green line, neighborhood evidence; Blue line, co-occurrence evidence; Purple line, experimental evidence; Yellow line, text mining evidence; Light blue line, database evidence; Black line, co-expression evidence. (A) Human, (B) Gorilla, (C) Chimpanzee, (D) Monkey, (E) Rabbit, (F) Rat, (G) Mouse, (H) Pig, (I) Sheep, (J) Buffalo, (K) Yak, (L) Goat and (M) Cattle.

across species (Fig. 7). Generally, we found that there were different proteins that clustered with CD14 in all the species. All species had 10 proteins in their cluster except cattle and goat that had 11. Looking at the Venn diagram, rabbit had the highest CD14 PPI that is not shared with others while three protein set (CD14, TLR2, and TLR4) is common to members of this group (Fig. 8A). Figure 8B shows the ruminant group, including goat, sheep, and yak had no unique gene set, meaning the PPI is duplicated in one or two other members of the group. However, cattle has eight unique PPI while buffalo has four that were not shared with others. CD14 and TLR2 are common to all in this group. Likewise, there were eight unique PPI in human, six in gorilla and none in monkey and chimpanzee (Fig. 8C).

## DISCUSSION

Comparative analysis of CD14 protein in this study enhances our understanding of genome plasticity among 14 mammalian species and establishes functional, molecular, and structural relationships in different clades that are important in an evolutionary trace. The significant variability in the MSA of the CD14 molecule across the species suggests a

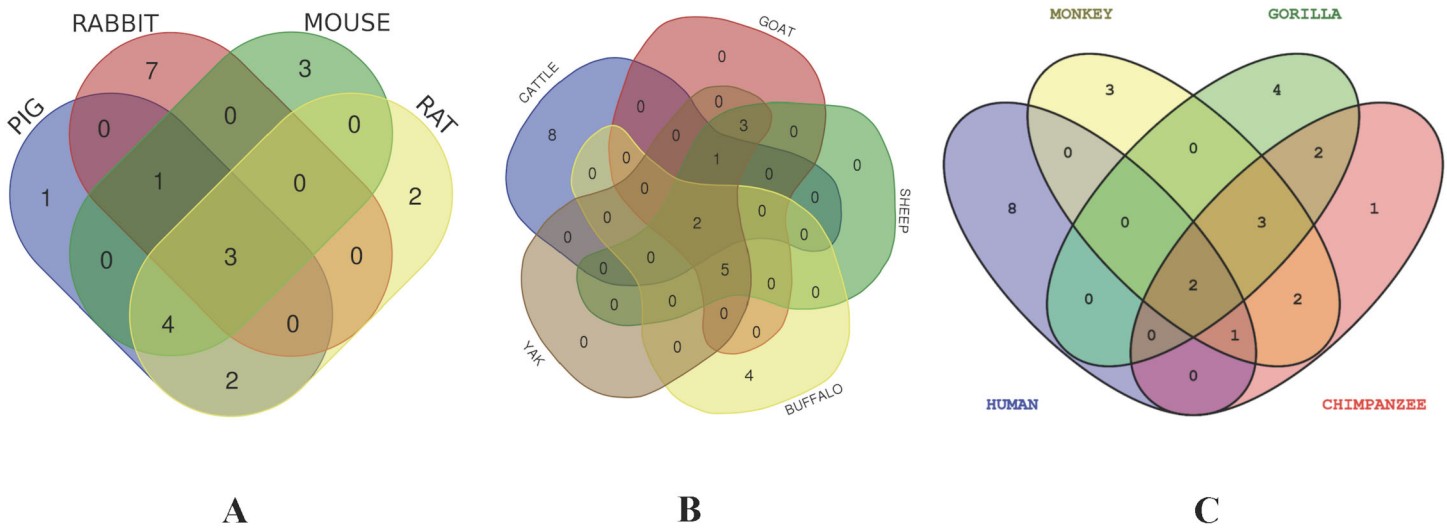

**Figure 8 Venn diagram showing the proportion of intersection and unique genes depicting evolutionary diversity of CD14 molecule.**
(A) Comparison and visualization of protein interaction with CD14 molecule in pig, rabbit, mouse, and rat. (B) Comparison and visualization of protein interaction with CD14 molecule in cattle, yak, sheep, goat, and buffalo. (C) Comparison and visualization of protein interaction with CD14 molecule in human, gorilla, chimpanzee, and monkey.

high evolutionary divergence especially between the ruminant and non-ruminant group. This implies that CD14 amino acid sequence had undergone significant changes during speciation leading to functional and structural modification in different species. Studies have shown that variation in amino acid sequences could impact immunogenicity, immunotolerance, and immunoreactivity (*Tauber et al., 2004*; *Kanduc, 2012*; *Bendl et al., 2014*). However, we found that amino acid residues like leucine (L), glutamic acid (E), lysine (K), valine (V), aspartic acid (D), glycine (G), serine (S), and asparagine (N) are highly conserved, thereby retaining some degree of homology in functional, molecular, and structural characteristics. In addition, this reveals the common origin between the mammalian species before divergent speciation (*Tauber et al., 2007*). Based on the percentage identity and similarity, monkey, gorilla, and chimpanzee are closer to human in their CD14 amino acid sequence, suggesting a lower degree of variation and this may infer some degree of similar CD14 expression during disease condition (*Ferrero et al., 1990*; *Ibeagha-Awemu et al., 2008*; *Bendl et al., 2014*).

We also observed that the molecular weight, isoelectric point, instability index and net charge of CD14 protein for this group of mammals are similar, suggesting a key biochemical and immunological function is retained in these species during evolution (*Saha et al., 2013*; *Ajayi et al., 2018*). Of interest, the CD14 sequence in cattle and buffalo were much more conserved than yak, despite their common origin potentially implying that domestication has not affected key biological functions in cattle, and the possibility that buffalo can also be domesticated without loss of immunological function. Furthermore, a higher aliphatic index, net negative charge and GRAVY as shown in the physicochemical properties of CD14 protein in mouse and rat gives an indication of high concentration of alanine, valine, isoleucine and leucine, reported to influence transcription factors, providing higher tolerance against bacterial, and viral infections (*Korber, 2000*; *Panaro et al., 2008*;

*Ivanov et al., 2015*). This is thought to be an important evolutionary adaptation for these small animals to survive bouts of exposure to diseases in their environment, and may explain the basis for these organisms at times serving as reservoir hosts for many disease pathogens in humans. The general negative net charge of CD14 protein as observed across the species indicates an increasing reactivity and help in its receptor binding mechanism. Therefore, the higher the net charge, the more the reactivity of the protein.

Interestingly, our motif and signal peptide scan found just one domain and one signal peptide in the entire length of the CD14 amino acid sequence. The numbers of conserved LRR domains vary from species to species. Species with similar number of LRR profile may likely have same immunological implications. This again, is a significant evolutionary signature. CD14 is a co-receptor that bind with LPS, therefore a higher leucine amino acid profile in the molecule may accelerate its binding mechanism to receptor in a significant way because the protein plays a significant regulatory role in initiating a robust innate immune response. Studies have shown that LRR domain is evolutionarily conserved in most of the innate immune related proteins in vertebrates, invertebrates and plants, providing the innate immune defense especially through pathogen-associated molecular patterns (*Ng & Xavier, 2011*). Some reports also stated that there about 2–45 LRRs within the LRR domains, containing up to 30 residues. Classifying our mammalian species under study into ruminants vs. non-ruminants, we observed that non-ruminants possess a lower number of LRR domain in their CD14 molecule (one domain in human, three in ruminants, and four in rat). Notably, rat again possesses the highest number of LRR domains remarkably traceable to selection pressure across the species. Moreover, the amino acid sequence of this domain is highly conserved for all species under study, and are found toward the C-terminal region of CD14, justifying the fact that amino acid sequence variation that differentiate species are found close to the N-terminal region (*Peters et al., 2018*).

Our study additionally reveals varying secretory signal peptide sites in the CD14 molecule across the species. Signal peptides have been identified as hydrophobic amino acids, recognized by the signal recognition particle in the cytosol of eukaryotic cells (*Dultz et al., 2008*). Secretory signal peptide is a class of signal peptide that allows the export of a protein from the cytosol into the secretory pathway (*Nielsen & Krogh, 1998*; *Park & Kanehisa, 2003*; *Rivas & Fontanillo, 2010*; *Sigrist et al., 2010*). In this, we found that human, monkey, gorilla, and chimpanzee all have the same signal peptide site and position. Cattle, yak, sheep, and buffalo also share the same site and position whereas goat did not, confirming why goat is significantly distant to other ruminants in our phylogenetic construction. It is unclear if this is related to disease tolerance when compared to other species. However, we noted in our predicted neural network that the subcellular localization of CD14 protein goes from the extracellular through the intracellular and enters the secretory pathway for all the species, except sheep. In sheep, the subcellular localization begins from the nucleus through the mitochondrion, peroxisomal targeting signal and N-terminal sequences before it enters into the secretory pathway. This information may possess potential immunological consequences that will require further analysis and possibly an in vitro validation.

Of most importance, a higher proportion of the predicted mutations occupying the C-terminal region of CD14 protein show that they are closer to the active site and may have direct structural and functional effects on the protein thereby causing harmful disease phenotype or susceptibility (*Malm & Nilssen, 2008*). Studies have shown that the LRRs at the C-terminal region is required for responses to smooth lipopolysaccharide, whereas the variable region (290–375) has been found to be necessary for response to bacterial lipopolysaccharide (*Bella et al., 2008*; *Arnesen, 2011*; *Xue et al., 2012*, *2018*). Therefore, variation at this region might be traceable to varied exposure and responses to pathogens in the cause speciation.

We observed a higher proportion of deleterious mutational spots in human, monkey, gorilla, and chimpanzee occupying the same loci compared to ruminants and other species. This might suggest that the vital residue conservation at this region is due to selection pressure among these species and has been maintained over time possibly because of their role in evolution, resulting in similar biological and immunological function (*Feder & Mitchell-Olds, 2003*; *De Donato et al., 2017*; *Peters et al., 2018*). Therefore, a perturbation of the amino acid sequence at this region could affect the protein folding, ligand binding and other functions which might be lethal or regarded as disease-causing mutation in all mammals (*Choi et al., 2012*). Understanding the molecular variation in the region could help solve the challenge of Mendelian disease phenotypes. We recommend an in vitro study of this region in CD14 protein sequence to elucidate the molecular mechanism affecting functionality of this region. In all, three of these mutations have been characterized and verified in humans to cause disruption of active site and loss of protein activities (*Singh & Borbora, 2018*).

Furthermore, we used the STRING database to annotate CD14 protein network with other protein molecules that may have evolved together during speciation. Significantly, we found that the CD14 molecule selectively interacts with other proteins from species to species. For example, in cattle, the CD14 molecule interacts with eight other proteins, which are not shared with goat, sheep, and yak. In a similar vein, buffalo has four unique sets of protein that co-express with CD14 protein. Human and gorilla in their group has eight and six genes, respectively, that uniquely interact with CD14 protein, which are not found in monkey and chimpanzee. These protein interactions are possibly due to the specific molecular or biochemical changes that occur in CD14 protein during selection pressure in different species. This interactome is important to decipher molecular and biochemical mechanisms shaped by evolution, which may be useful for drug design and therapeutic treatment of many diseases. Several studies have shown that molecular association between chains of different protein molecules is geared by the electrostatic force like hydrophobic effects which define specific bimolecular interaction in different organism (*Arkin, Tang & Wells, 2014*; *De Las Rivas & Fontanillo, 2010*; *Chen, Krinsky & Long, 2013*). The modulation of this interaction may be useful as putative therapeutic targets for disease treatment in many species. *Ivanov et al. (2014)* have used the interaction of Tirobifan with glycoprotein IIb/IIIa as an inhibitor for cardiovascular drug discovery, likewise the interaction of Maraviroc and CCR5-gp120 for anti-HIV drug.

As shown earlier, there are variations in the number of the LRR domain among these species, possibly the lesser number of LRR domain in human is supplemented or accounted for by the functionality of other genes in the network (*Thakur & Shankar, 2016*). From our physicochemical properties, CD14 is classified as hydrophobic across the species due to higher proportion of LRR. The varying degree of LRR among these species is thought to affect the electrostatic force created by the hydrophobic effects of the protein. Published studies have shown that diverse fungal, bacterial, viral, and parasite components are sensed by the mammalian LRR domain of proteins like NOD-like receptors and Toll-like receptors (*Korber, 2000*; *Kutay & Güttinger, 2005*; *Lucchese et al., 2009*; *Kamaraj & Purohit, 2014*). Likewise, about 34 LRR proteins have been associated with diseases in human. Obviously, divergent evolutionary events have shaped the PPI of CD14 in different species, which is thought to be significant to varying degrees of disease susceptibility and pathogen selection.

## CONCLUSION

We have used computational methods to gather information on CD14 protein in 14 mammals. Our in silico comparison of CD14 amino acid sequences among these species gave molecular evidence of divergent evolutionary events that occurred during speciation, potentially of significance in modulating innate immune response to pathogenic challenges. Obviously, this gene has been subjected to selection pressure due to sufficient sequence variation we found from one species to another. We identified mutational hotspots with damaging effects in human and other species. In particular, the signal peptides located in these mutational hotspots are possibly of major importance in immunological studies. The variants identified in this study can be further subjected to validation through in vitro analysis. Since the CD14 molecule is essential in initiating proper immune response to pathogens and the precursor of a robust adaptive immune response, our study highlights the effect of mutations on protein structure and disease outcome, PPI that may be essential for drug design, yielding themselves to therapeutic manipulations for treating many diseases. Finally, these results contribute to our understanding of the evolutionary mechanism that underlie species variation in response to complex disease traits.

### Funding

This work was funded by a Laboratory and Faculty Development Award, College of Health Sciences and Technology, Rochester Institute of Technology (Bolaji Thomas). Olanrewaju Morenikeji is supported through the American Association of Immunologists Careers in Immunology Fellowship Program. The funders had no role in study design, data collection and analysis, decision to publish, or preparation of the manuscript.

### Grant Disclosures

The following grant information was disclosed by the authors:
Laboratory and Faculty Development Award, College of Health Sciences and Technology,

Rochester Institute of Technology.
American Association of Immunologists Careers in Immunology Fellowship Program.

## Competing Interests

The authors declare that they have no competing interests.

## Author Contributions

- Olanrewaju B. Morenikeji conceived and designed the experiments, performed the experiments, analyzed the data, contributed reagents/materials/analysis tools, prepared figures and/or tables, authored or reviewed drafts of the paper, approved the final draft.
- Bolaji N. Thomas conceived and designed the experiments, performed the experiments, analyzed the data, contributed reagents/materials/analysis tools, prepared figures and/or tables, authored or reviewed drafts of the paper, approved the final draft.

## Data Availability

Complete amino acids sequences mammalian species are available at UNIPROT: Human: P08571; Rat: Q63691; Mouse: P10810; Cattle: Q95122; Rabbit: Q28680; Monkey: B3Y6B8; Gorilla: G3R4C0; Sheep: W5QJA2; Horse: F6VK89; Pig: A7BG66; Buffalo: A0A2R4SDF9; Chimpanzee: B3Y6B4; Yak: L8I9P7. The amino sequence for goats is available from NCBI: ABE68725.1.

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
