# Peer review of "In silico analyses of CD14 molecule reveal significant evolutionary diversity, potentially associated with speciation and variable immune response in mammals"

_PeerJ, doi:10.7717/peerj.7325_

## Round 0.1 · original submission · Minor Revisions

The reviewers have appreciated the comprehensive analysis reported in the paper, but they raised minor concerns, especially in presentation and copy-editing, which should be addressed before the paper is acceptable for publication.

·

Basic reporting

No comments

Experimental design

No comments

Validity of the findings

No comments

Additional comments

The manuscript entitled ‘In silico analyses of CD14 molecule reveals significant evolutionary diversity potentially associated with speciation and variable immune response in mammals’ is well written and questions that are raised have been answered using various computational programs.
Following incorporation will improve the manuscript.
Table 1 and Table 2 show identity and similarity respectively, does not give much different interpretation/ information, should be merged in one table, keeping ‘similarity’ in bracket.
Author should construct a figure depict the gene structure indicating conserved domain, location of mutation/ substitution of amino acids or hotspots and whether deleterious or neutral etc.

Reviewer 2 ·

Basic reporting

- Overall, the paper is clearly written. A minor comment is that the word "the" is sometimes missing; e.g. *the* CD14 gene, *the* CD14 molecule, *the* CD14 protein, *the* evolutionary tree was constructed.
- I do not see a reason to abbreviate amino acids as aa on line 78 if you spell out the full word later (such as on lines 86, 115 and elsewhere).

Experimental design

- The authors layout the article in an easy to follow manner, and the provide an appropriate amount of detail in the methods section.
- I am curious as to why the particular 14 mammalian species were chosen?

Validity of the findings

Data is robust, and conclusions are supported by the results, but the authors should make the limitations of their study more clear in the Discussion.

Additional comments

- It would be nice to briefly define the amino acids properties listed on lines 114-118.
- I have also made a few minor comments in the PDF attached.
- Tables 1 and 2 could be easier to read if there was a colour-scale (in addition to the raw numbers) signifying the percents. However, if there are colour-charges involved, then this comment can be ignored.
- Is there a rationale for the row order in Tables 3,4,5? If not, there should be? Even if the reason is simply alphabetical.
- Figures 1 and 7 are too small/difficult to read.

Annotated reviews are not available for download in order to protect the identity of reviewers who chose to remain anonymous.

Reviewer 3 ·

Basic reporting

The article needs some English Language editing and i have identified and also suggested how to improve the wrong phrases
Line 26 -replace report with paper
Line 30 - replace ancient conservation with is conserved
Line 33 - replace this is important with this may be important
Line 63 is a repetition of line 50
Line 77 - change gather to glean
The objective statement is not obvious from this introduction. Authors should include the objective statement
Line 171-172 should be rephrased to reflect that CD14 protein sequence is not conserved
Line 176 - change show to shows
Line 249 - change eleven to 11
For the discussion section of this paper, there are many statements made by the authors that needs to be supported by citations
Line 273 - Citation should be included after disease condition
Line 278 - recast the statement because it appears something is missing here
Line 284 - Citation be included after viral infections
Line 293 - The number and conservation of LRR and delete the conservation of LRR domain in line 294
Line 316 - Citation should be included after pathway
Line 319- insert when between tolerance and compared
Line 327- Recast the statement here
Line 332 - Insert a citation after lipopolysaccharide
Line 347 - Citations should be behind protein activities
Genes and proteins were used interchangeably in this paper, authors needs to be more careful and use the terms where appropriate
Line 357 - change current statement to is important to decipher molecular and biochemical mechanism
Every where the authors talked about evolutionary event, they need to indicate whether it is divergent, convent or even neutral evolution

Overall, the authors needs to indicate that CD14 is a gene that encodes a protein and clear any confusion from the use of the terms interchangeably.

Experimental design

No comment

Validity of the findings

The authors should use the word may where they speculated in this paper.

Additional comments

See comments above

---

## Round 0.2 · accepted · Accept

You have sufficiently addressed the comments from the reviewers. I believe your study of CD14 and its evolution within mammals will add to our understanding of evolution of immune response to pathogens.